# Coronary heart disease and stroke mortality trends in Brazil 2000–2018

**Patrícia Vasconcelos Leitão Moreira**[1][*], **Adélia da Costa Pereira de Arruda Neta**[1], **Sara Silva Ferreira**[1], **Flávia Emília Leite Lima Ferreira**[1], **Rafaela Lira Formiga Cavalcanti de Lima**[1‡], **Rodrigo Pinheiro de Toledo Vianna**[1‡], **Jevuks Matheus de Araújo**[2‡], **Rômulo Eufrosino de Alencar Rodrigues**[2‡], **José Moreira da Silva Neto**[3‡], **Martin O'Flaherty**[4]

1 Department of Nutrition, Federal University of Paraiba, João Pessoa, Paraíba, Brazil, 2 Department of Economy, Federal University of Paraiba, João Pessoa, Paraíba, Brazil, 3 Technical School of Health of the Federal University of Paraíba, Joao Pessoa, Paraíba, Brazil, 4 Department of Public Health and Policy, University of Liverpool, Liverpool, Merseyside, United Kingdom

☯ These authors contributed equally to this work.
‡ These authors also contributed equally to this work.
* patricia.moreira@academico.ufpb.br, patriciamoreira1111@hotmail.com

**Data Availability Statement:** All relevant data are within the paper.

## Abstract

### Objective

To analyse the mortality rate trend due to coronary heart disease (CHD) and stroke in the adult population in Brazil.

### Methods

From 2000 to 2018, a time trend study with joinpoint regression was conducted among Brazilian men and women aged 35 years and over. Age-adjusted and age, sex specific CHD and stroke trend rate mortality were measured.

### Results

Crude mortality rates from CHD decreased in both sexes and in all age groups, except for males over 85 years old with an increase of 1.78%. The most accentuated declining occurred for age range 35 to 44 years for both men (52.1%) and women (53.2%) due to stroke and in men (33%) due to CHD, and among women (32%) aged 65 to 74 years due to CHD. Age-adjusted mortality rates for CHD and stroke decreased in both sexes, in the period from 2000 to 2018. The average annual rate for CHD went from 97.09 during 2000–2008 to 78.75 during 2016–2018, whereas the highest percentage of change was observed during 2008 to 2013 (APC -2.5%; 95% CI). The average annual rate for stroke decreased from 104.96 to 69.93, between 2000–2008 and 2016–2018, and the highest percentage of change occurred during the periods from 2008 to 2013 and 2016 to 2018 (APC 4.7%; 95% CI).

### Conclusion

The downward trend CHD and stroke mortality rates is continuing. Policy intervention directed to strengthen care provision and improve population diets and lifestyles might explain the continued progress, but there is no room for complacency.

**Funding:** The National Council for Scientific and Technological Development (Conselho Nacional de Desenvolvimento Científico e Tecnológico – CNPq) supported all the funding or sources of support received during this study; Grant number 442891/2019-9. The funders had no role in study design, data collection and analysis, decision to publish, or preparation of the manuscript.

**Competing interests:** The authors have declared that no competing interests exist.

## Introduction

Cardiovascular diseases (CVD) represent a significant public health problem, because they are the leading cause of death and disability that affects adults at full productive age, accounting for 31% of global deaths [1, 2] and about 27.7% of deaths in Brazil [3]. These diseases have an enormous economic impact on the country, resulting in losses of potential years of life and a high burden on the public health system, mainly in cost to the state [4–7]. With increasing longevity and the relative decline of infectious diseases, population ageing contributes to increased CVD burden [4, 8], alongside persistent socioeconomic inequalities in risk factors and CVD death [9, 10].

Some studies have already analysed cardiovascular diseases mortality trends in Brazil, over longer time frames in the 90s [11], or focusing in large cities [12], from 1990 to 2009 [1] and 1980 to 2012 [13]. They showed that CVD trends are characterized by a drop in mortality in more advanced age groups, resulting from the delay in deaths caused by chronic diseases [13].

However, none of these studies have focused on identifying periods of similar annual rate change in an unbiased way despite many studies. The only existing study that used a similar methodology analysed the period 2000–2015, but focused only in the most populous cities, not the entire country [14].

In several developed countries, slowdowns in CVD mortality trends have been reported, and likely explaining the smaller gains in life expectancy experienced recently [15]. Describing current CVD mortality trends is essential to continuously assess the disease's burden and understand the evolution of the epidemiological transition in Brazil. This study aims to analyse the mortality rates trends due to CVD (coronary heart disease and stroke) in the adult population in Brazil from 2000 to 2018.

## Materials and methods

### Mortality and population data

Mortality rates due to CVD were analysed in Brazil, from 2000 to 2018. Data on mortality were obtained through the Unified Health System database *(Base de dados do Sistema Único de Saúde—DATASUS)*, from the Ministry of Health of Brazil [3]. Population data were obtained from the database of the Brazilian Institute of Geography and Statistics *(IBGE)* [16]. The causes of mortality were classified according to the 10th revision of the International Classification of Diseases—ICD 10, with coronary heart diseases (CHD) grouped in codes from I20-I25, and stroke in codes I60-I69 [17]. Age ranges were grouped according to the Pan American Health Organization standard, and data were obtained for both sexes: 25–34 years, 35–44 years, 55–64 years, 65–74 years, 75–84 years and >85 years.

Crude mortality rate was estimated by dividing the number of deaths from each disease by the number of individuals in the respective age group per 1000 people, per year [18]. Standardised mortality rate was calculated through the direct method, using the standard population from Brazilian Institute of Geography and Statistics *(IBGE)*—Census of 2010 [16].

The research protocol was approved by The Ethics Committee of Federal University of Paraíba with consent number 3.843.739. As these are secondary data made available on public domain sites of the Brazilian Unified Health System, ethics committee waived the requirement for informed consent.

### Trend analyses

To analyze the mortality rates over time an identity in an unbiased way periods of constant rate change and specific years when significant change in the trend occur, we used joinpoint

regression for estimation of the Annual Percentage Change (APC) [19]. The APC is one way to characterise trends in mortality rates over time, and it was estimated by fitting a regression line to the natural logarithm of the rates and using the calendar year as an independent variable. Several studies have used this methodology to estimate time trends in mortality rates [20–22]. We used the Grid Search's Methods and a Bayesian Information Criterion (BIC) approach to select the model that best fitted the data [19].

## Results

Fig 1 shows data on the trend of mortality in Brazil from 2000 to 2018 in the general population and by sex for CHD and stroke.

Between 2000 and 2018, crude mortality rates from CHD decreased in both sexes and in all age groups, except for males, which increased by 1.78% in the age group over 85 years of age (Fig 2). The most pronounced decline was observed in males aged 35 to 44 years (33%) and, in females, aged 65 to 74 years (32%) (Fig 2). The gross mortality rates for stroke, between 2000 and 2018, also decreased in both sexes and age groups, with no exceptions (Fig 3). The age group between 35 and 44 years had the most accentuated decline, both in males (52.1%) and in females (53.2%) (Fig 3).

Mortality rates for CHD adjusted for age decreased in both sexes, in the period from 2000 to 2018. The average annual rate went from 97.09 in the period from 2000 to 2008 to 78.75 in the period from 2016 to 2018, its highest percentage of change observed in the period from 2008 to 2013 (APC -2.5%; 95% CI). The average annual rate for males went from 115.89 to 97.23 between the periods 2000–2008 to 2016–2018, with the highest percentage of change observed between 2008 and 2013 (APC -2.3%; 95% CI). The average annual rate for females went from 79.82, in the period from 2000 to 2008, to 61.91 in the period from 2016 to 2018, with the highest percentage of change observed in the period from 2008 to 2012 (APC 3.0%; 95% CI).

There was also a decrease in age-adjusted mortality rates for stroke in both sexes, for the same period (2000–2018). The average annual rate decreased from 104.96 to 69.93, between the periods 2000–2008 and 2016–2018. For this same period, the average annual rate decreased from 110.3 to 74.51 for males and from 100.16 to 65.74 for females. The highest percentages of

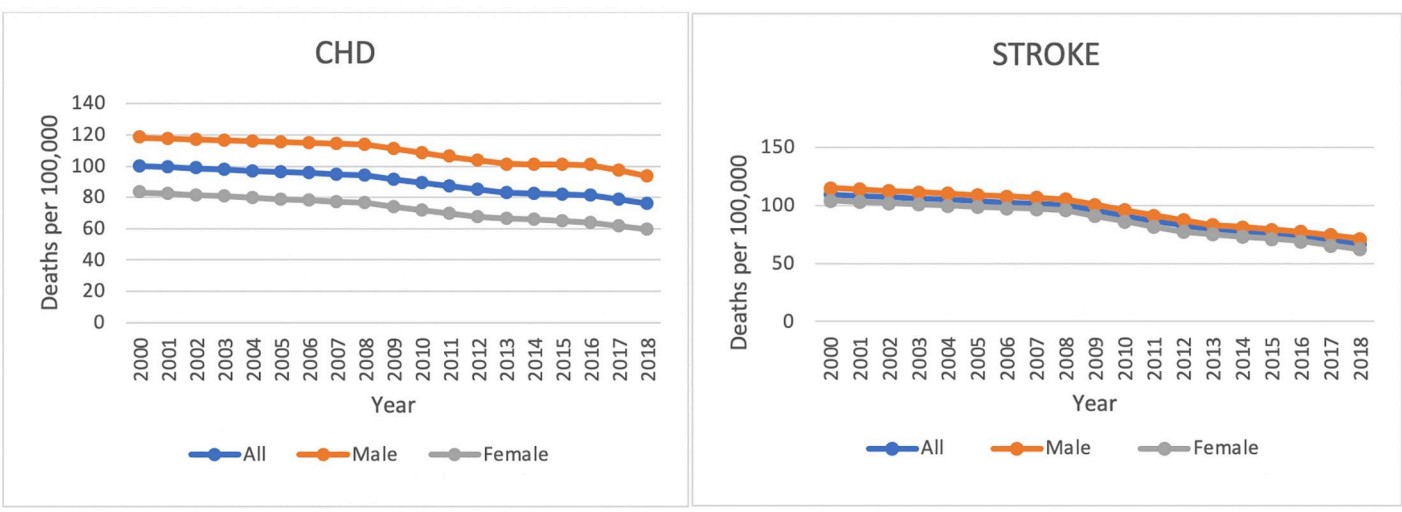

**Fig 1. Trends in age-standardized mortality rates per 100,000 by sex for coronary heart disease and stroke.** Brazil. 2000–2018.

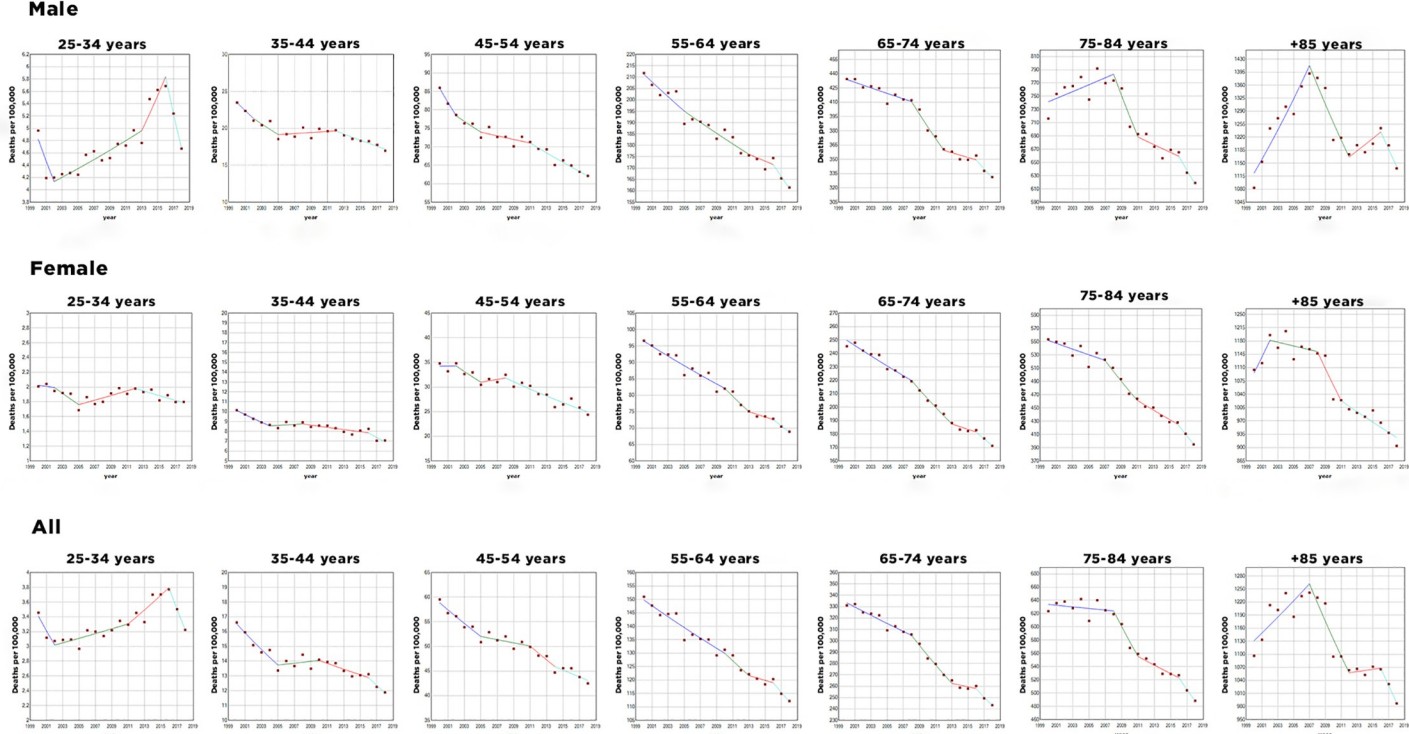

**Fig 2. Trends in age and sex-specific mortality rates per 100,000 for coronary heart disease, Brazil.** 2000–2018.

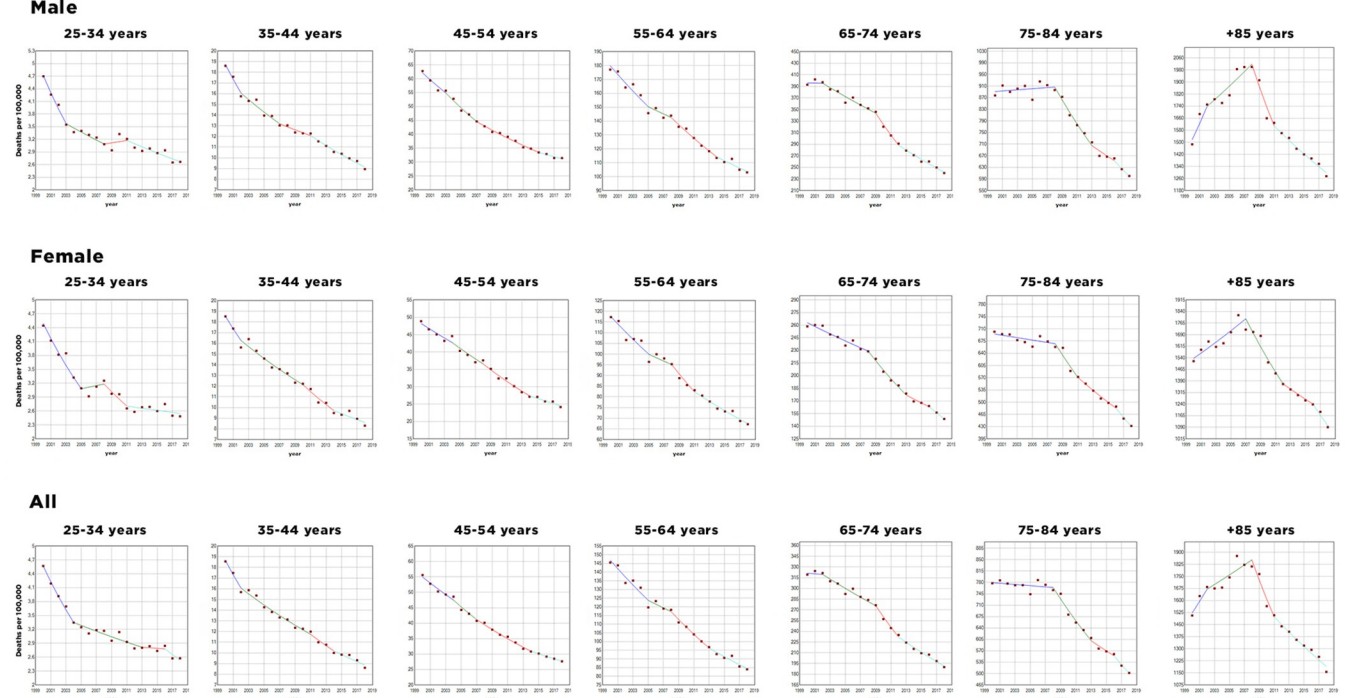

**Fig 3. Trends in age and sex-specific mortality rates per 100,000 for stroke, Brazil.** 2000–2018.

change were in the period from 2008 to 2013, for males (APC -4.6%; 95% CI), and for females in the periods 2008 to 2012 and 2016 to 2018 (APC -5.2%; 95% CI). The highest percentage of general change was in the periods from 2008 to 2013 and 2016 to 2018 (APC 4.7%; 95% CI) (Table 1).

## Discussion

This study aimed to investigate the trend in CHD and stroke mortality rate in the Brazilian adult population from 2000 to 2018. We describe a continuing decline in crude and adjusted mortality rates for CHD and stroke in both sexes and in all age groups, except for CHD in males aged older than 85 years, the most accentuated declining occurred for age range 35 to 44 years for both sexes due to stroke and in males due to CHD. An important decline was observed in females aged 65 to 74 years for CHD.

The results found in this study is consistent with other studies done in Brazil. Martins *et al.* [14] investigated trends in mortality rate from cardiovascular disease and cancer between 2000 and 2015 in five capitals cities of the five regions in Brazil and found a consistent decrease in mortality rate from CVD in all capital cities for all age and sexes, except for males aged 70 years and older in Manaus. Mansur and Favarato [1] found a decrease in mortality rate due to CVD, CHD and stroke in males and females during the period of 1980–2012, however between 2007 and 2012 there was not significative differences in mortality due to CHD; when investigating differences between males and females, statistical difference was found only during the period of 1980–2006 with males having a higher decrease in mortality due to CHD.

Garritano *et al.* [23] evaluated the mortality trends due to stroke in Brazil between 2000 and 2009 and found an increase in mortality until 2006 followed by a decrease until 2009. They observed a tendency to decline in mortality rate in both males (-14,69%) and females (-17%) and -14.99% in total, the age range 30–49 years showed a tendency of continue e linear decreasing in mortality rate. Despite the increase in the absolute number of deaths, Lotufo *et al.* [24] found a decrease in mortality due to stroke in Brazil from 1980 to 2015 in those younger than 70 years old, mostly accentuated in females.

**Table 1. Age-standardized CHD and stroke mortality rates per 100,000.** Brazil 2000–2018.

| Sex | CHD | | | | Stroke | | | |
|---|---|---|---|---|---|---|---|---|
| | Period | AMR | Mortality rates (min-max) | APC | Period | AMR | Mortality rates (min-max) | APC |
| **All** | 2000–2008 | 97,09 | 94,13–100,11 | -0.8* | 2000–2008 | 104,96 | 100,37–109,67 | -1.1* |
| | 2008–2013 | 88,45 | 82,96–94,13 | -2.5* | 2008–2013 | 89,23 | 78,79–100,37 | -4.7* |
| | 2013–2016 | 82,28 | 82,28–81,61 | -0.5 | 2013–2016 | 76,02 | 73,3–78,79 | -2.4 |
| | 2016–2018 | 78,75 | 75,94–81,61 | -3.5 | 2016–2018 | 69,93 | 66,63–73,30 | -4.7* |
| **Male** | 2000–2008 | 115,89 | 113,62–118,19 | -0.5* | 2000–2008 | 110,30 | 105,59–115,14 | -1.1* |
| | 2008–2013 | 107,93 | 101,36–113,62 | -2.3* | 2008–2013 | 94,13 | 83,37–105,59 | -4.6* |
| | 2013–2016 | 100,99 | 100,63–101,36 | -0.2 | 2013–2016 | 80,42 | 77,53–83,37 | -2.4* |
| | 2016–2018 | 97,23 | 93,88–100,63 | -3.4 | 2016–2018 | 74,51 | 71,54–77,53 | -3.9* |
| **Female** | 2000–2008 | 79,82 | 76,50–83,22 | -1.0* | 2000–2008 | 100,16 | 96,04–104,38 | -1.0* |
| | 2008–2012 | 67,61 | 67,61–76,50 | -3.0* | 2008–2012 | 86,60 | 77,65–96,04 | -5.2* |
| | 2012–2016 | 65,87 | 64,16–67,61 | -1.3 | 2012–2016 | 73,41 | 69,29–77,65 | -2.8* |
| | 2016–2018 | 61,91 | 59,69–64,16 | -3.5 | 2016–2018 | 65,74 | 62,26–69,29 | -5.2* |

Periods were identified by Joinpoint Regression Analysis

* p-value < 0.01

[a] AMR Average mortality rate

[b] APC Annual Percent Change.

Our study confirms that CHD and stroke mortality has been on the decline in the country since 2000, with projections of a further decline by 2030 [25]. Data show that cardiovascular diseases were responsible for most deaths in the country at the end of the last century, mainly of people over 60 years old [25]. However, despite population ageing, the general mortality in this population has decreased, with the increasing probability of a person who in the year 2000 was sixty reaching the ninety 30 years later. The decline in mortality from cardiovascular disease, especially in these age groups, contributes to a new epidemiological picture in the country [25], and likely contributing to the increase in life expectancy [15].

Since the emergence of the Unified National Health System (*Sistema Único de Saúde—SUS*), the number of people seeking a primary health care service has increased by 450% until 2008 [14, 26]. The Family Health Program (*Programa de Saúde da Família—PSF*), created in 1994, has contributed in recent years to a 44% reduction for all causes of mortality in the urban population in a situation of social vulnerability in Rio de Janeiro, with cardiovascular diseases being those that obtained one of the more significant absolute reductions in the number of deaths. It is important to highlight that the service users had a lower mortality rate than non-users, which shows the importance of the health service in this premise [27].

The prevention of CHD is a challenge owing to the complexity of the factors involved in the physiopathology of atherosclerosis, while the control of systemic arterial hypertension has important impact on cerebrovascular diseases such as stroke, so much factors is involved in the diagnosis and treatment of CHD, i.e. dyslipidaemia, smoking and diabetes, commonly unknown until the first coronary event [1]. Reflecting that the management of risk factors for these diseases is crucial for reducing mortality, and that hypertension is the main risk factor for stroke, it is important to highlight that the primary care has a system for registering and monitoring hypertensive and diabetic patients called HIPERDIA (acrostic for hypertension and diabetes), implemented in 2002, which aims to manage the distribution of medicines to this population. Lopes *et al.* [28] reported a drop in hospitalizations for stroke and, consequently, a drop in mortality, which was linked, among other factors, to the implementation of this program in Brazil.

Our study confirms that CVD mortality trends in Brazil are similar to those countries experiencing sustained declines of CVD since the last part of the 20th century. In several countries in Europe reductions in CHD or stroke mortality has been reported, such as England [29, 30], Netherlands [22], Sweden [31], Scotland [20] and Germany [32]. In Latin America, the picture is more complex. Countries such as Argentina and Colombia showed declines of 51% and 6,5%, respectively, while Mexico showed an increase of 61% in mortality rate [21].

However, alarming trends are being observed in developed countries. This downward trend that has been observed since the mid-1970s has slowed down, and the life expectancy for some countries' population has decreased considerably. Most likely, this may be associated with an increase in the prevalence of obesity and diabetes [15]. It was observed, in some European countries and in the United States, a slowdown or stagnation of improvements on CVD mortality [33, 34]. As in Mexico, this recent trend might be linked to raises in obesity and diabetes, rather than deterioration of health services.

In our study most pronounced decline in deaths from CHD and stroke occurs as of 2008, regardless of sex, which culminates in the beginning of flu vaccination campaigns in Brazil, which began in 2009 [35]. Bacurau *et al.* [36] observed that mortality from cardiovascular events and cerebrovascular diseases showed an important reduction with the beginning of vaccination campaigns against influenza in the elderly. The risk of the influenza virus evolving to cardiac events, especially in elderly populations, is already well defined in the literature, but the effectiveness of influenza vaccination in reducing mortality from cardiovascular diseases is

still controversial. Nevertheless, systematic reviews of longitudinal and randomized clinical studies support the hypothesis of an important relationship between these two events [37–39].

However, the decreasing trend of CHD and stroke deaths in Brazil cannot give us the reassurance that the problem is under control. Brazil still has high mortality rates from these diseases and remains the largest cause of deaths in the country [40] and in the world [41, 42]. While the improvement of some factors such as health units of primary care, vaccination campaigns, and the Unified National Health System's distribution of medicines, these gains can be lost because of the rise of the diabetes and obesity burden. Future projections for diabetes in Brazil suggest an increase in 144% of cases by the year 2040, with premature mortality from CDV attributed directly to diabetes causing 31.6% of the 2017 burden [43]. Mortality and disability-adjusted life years (DALYs) lost to non-communicable diseases caused by high body mass index (BMI) was observed in 2017. High BMI was responsible for 12.3% of all deaths and 8.4% of total DALYs lost to non-communicable diseases [44].

Public policies aimed at the elderly and for the control of risk factors for CVD need to be strengthened. Some actions that could lead to a decrease in mortality rate are the control of risk factors, prevention of circulatory diseases in primary and secondary care levels, an improvement in the population socioeconomic status, an investment in high technology procedures as well an enlargement in the amount of equipment for accurate diagnosis and faster attendance could lead to a decreasing in mortality [23]. Promoting healthier diets, physical activity, reducing smoking and tackling obesity and diabetes continue to be public health priorities. In this sense, policy actions such as the Food Guide for the Brazilian Population has been widely disseminated [45], as well as the academies in the squares known as free Outdoors Academies (*Academia ao ar livre—AAL*) are implemented by city halls across the country.

This study is the first national study in Brazil to be developed using an unbiased methodology to describe the trend. Several previous studies have been published in the country so far [1, 12, 13, 46], but they have not analysed the inflection points using this methodology.

Some important limitations must be mentioned. Like any other study that uses mortality data across multiple versions of the International Classification of Diseases (ICD), there is potential for attribution bias owing to both the change between versions of ICD and the procedures used to code deaths. Vital statistics come from an independent registry called Unified Health System *(DATASUS)* and are subject to errors. As well as the projection data of the Brazilian population in relation to the last demographic census are delayed being released because of the pandemic involving the coronavirus. Thus, the last census released in the country was carried out in 2010 and it was used in the study.

Given the country's vast expanse, a specific analysis by the regions of the country would be useful to further explore if the declines are generally occurring across populations with different levels of socioeconomic status.

## Conclusions

The downward trend CHD and stroke mortality rates is continuing. Policy intervention directed to strengthen care provision and improve population diets and lifestyles might explain the continued progress, but there is no room for complacency.

## Author Contributions

**Conceptualization:** Patrícia Vasconcelos Leitão Moreira, Adélia da Costa Pereira de Arruda Neta, Flávia Emília Leite Lima Ferreira, Rafaela Lira Formiga Cavalcanti de Lima, Rodrigo Pinheiro de Toledo Vianna, Jevuks Matheus de Araújo, José Moreira da Silva Neto.

**Data curation:** Adélia da Costa Pereira de Arruda Neta, Jevuks Matheus de Araújo, Rômulo Eufrosino de Alencar Rodrigues.

**Formal analysis:** Patrícia Vasconcelos Leitão Moreira, Adélia da Costa Pereira de Arruda Neta, Jevuks Matheus de Araújo, Rômulo Eufrosino de Alencar Rodrigues, Martin O'Flaherty.

**Funding acquisition:** Patrícia Vasconcelos Leitão Moreira.

**Methodology:** Patrícia Vasconcelos Leitão Moreira, Flávia Emília Leite Lima Ferreira, Jevuks Matheus de Araújo, Rômulo Eufrosino de Alencar Rodrigues, Martin O'Flaherty.

**Project administration:** Patrícia Vasconcelos Leitão Moreira.

**Supervision:** Patrícia Vasconcelos Leitão Moreira.

**Writing – original draft:** Patrícia Vasconcelos Leitão Moreira, Adélia da Costa Pereira de Arruda Neta, Sara Silva Ferreira, Flávia Emília Leite Lima Ferreira, Rafaela Lira Formiga Cavalcanti de Lima, Martin O'Flaherty.

**Writing – review & editing:** Patrícia Vasconcelos Leitão Moreira, Adélia da Costa Pereira de Arruda Neta, Sara Silva Ferreira, Flávia Emília Leite Lima Ferreira, Rafaela Lira Formiga Cavalcanti de Lima, Rodrigo Pinheiro de Toledo Vianna, José Moreira da Silva Neto, Martin O'Flaherty.

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
