## [Decision Letter · Decision Letter 0]

12 May 2021

PONE-D-21-08855

Coronary Heart Disease and stroke mortality trends in Brazil 2000-2018

PLOS ONE

Dear Dr. MOREIRA,

Thank you for submitting your manuscript to PLOS ONE. After careful consideration, we feel that it has merit but does not fully meet PLOS ONE’s publication criteria as it currently stands. Therefore, we invite you to submit a revised version of the manuscript that addresses the points raised during the review process.

We look forward to receiving your revised manuscript.

Kind regards,

Venkata Naga Srikanth Garikipati, PhD

Academic Editor

PLOS ONE

Journal Requirements:

3) Please provide additional details regarding participant consent. In the ethics statement in the Methods and online submission information, please ensure that you have specified (1) whether consent was informed and (2) what type you obtained (for instance, written or verbal, and if verbal, how it was documented and witnessed). If your study included minors, state whether you obtained consent from parents or guardians. If the need for consent was waived by the ethics committee, please include this information.

4) Thank you for stating in your Funding Statement:

 [This work was partially supported by Conselho Nacional de Desenvolvimento

Científico e Tecnológico – CNPq; Grant number 442891/2019-9. Agency that granted a

scholarship and resources for the purchase of equipment.]. 

5) Please upload a new copy of Figures 2 & 3 as the detail is not clear. Please follow the link for more information: https://blogs.plos.org/plos/2019/06/looking-good-tips-for-creating-your-plos-figures-graphics/" https://blogs.plos.org/plos/2019/06/looking-good-tips-for-creating-your-plos-figures-graphics/

Reviewers' comments:

Reviewer's Responses to Questions

**Comments to the Author**

1. Is the manuscript technically sound, and do the data support the conclusions?

Reviewer #1: Yes

Reviewer #2: No

2. Has the statistical analysis been performed appropriately and rigorously? 

Reviewer #1: Yes

Reviewer #2: No

3. Have the authors made all data underlying the findings in their manuscript fully available?

Reviewer #1: Yes

Reviewer #2: No

4. Is the manuscript presented in an intelligible fashion and written in standard English?

Reviewer #1: Yes

Reviewer #2: No

5. Review Comments to the Author

Reviewer #1: The authors have written the paper technically sound, written in good English, statistical analysis are also well performed. The authors aim to understand the Coronary Heart Disease and stroke mortality trends in Brazil 2000-2018. I personally think are successful in addressing the issue.

Reviewer #2: The manuscript PONE-D-21-08855, titled “Coronary Heart Disease and stroke mortality trends in Brazil 2000-2018” by Patricia et al., they have analysed the trend of mortality due to coronary heart disease and stroke in the adult population in Brazil.

They have reported the decrease in crude mortality rates in both sexes and all age groups with exception in males over 85 years where the rate was increased. The declining trend was observed in males in age group 35 to 44 years and females in 65-74 years. The gross mortality for strokes was also decreased in both sexes and age groups between 2000 and 2018, with no exceptions. But this paper cannot be accepted under Research article category as the data is not sufficient and does not match the standard of PLOS One.

6. PLOS authors have the option to publish the peer review history of their article (what does this mean?). If published, this will include your full peer review and any attached files.

Reviewer #1: No

Reviewer #2: No

---

## [Author Response · Author response to Decision Letter 0]

20 May 2021

Reviewer #1

Response: Thank you for these constructive comments.

Reviewer #2

Response: #1. Reviewer 2 is unhappy with data not being enough. We will be grateful if the reviewer can elaborate more and indicate the purposes and how the data is inadequate, we can provide a better response. 

#2. The reviewer then states that the manuscript should not be accepted as a Research Article, as it doesn't comply with the journal standards. We politely disagree with the reviewer. For example., our paper describes the use of join point regression analysis on national mortality data in a similar way as we and others discussed trends in other settings (see for example https://journals.plos.org/plosone/article?id=10.1371/journal.pone.0114027

or https://journals.plos.org/plosone/article?id=10.1371/journal.pone.0059608#:~:text=Between%201982%20and%202006%2C%20the,year)%20(Table%201). We kindly ask the reviewer to specify which aspects do not fit the criteria of the journal, as it will enable us to address the points more precisely.

---

## [Decision Letter · Decision Letter 1]

10 Jun 2021

Coronary Heart Disease and stroke mortality trends in Brazil 2000-2018

PONE-D-21-08855R1

Dear Dr. MOREIRA,

We’re pleased to inform you that your manuscript has been judged scientifically suitable for publication and will be formally accepted for publication once it meets all outstanding technical requirements.

Kind regards,

Venkata Naga Srikanth Garikipati, PhD

Academic Editor

PLOS ONE

Additional Editor Comments (optional):

Reviewers' comments:

Reviewer's Responses to Questions

**Comments to the Author**

1. If the authors have adequately addressed your comments raised in a previous round of review and you feel that this manuscript is now acceptable for publication, you may indicate that here to bypass the “Comments to the Author” section, enter your conflict of interest statement in the “Confidential to Editor” section, and submit your "Accept" recommendation.

Reviewer #1: All comments have been addressed

Reviewer #2: All comments have been addressed

2. Is the manuscript technically sound, and do the data support the conclusions?

Reviewer #1: Yes

Reviewer #2: Partly

3. Has the statistical analysis been performed appropriately and rigorously? 

Reviewer #1: Yes

Reviewer #2: I Don't Know

4. Have the authors made all data underlying the findings in their manuscript fully available?

Reviewer #1: Yes

Reviewer #2: Yes

5. Is the manuscript presented in an intelligible fashion and written in standard English?

Reviewer #1: Yes

Reviewer #2: Yes

6. Review Comments to the Author

Reviewer #1: (No Response)

Reviewer #2: (No Response)

7. PLOS authors have the option to publish the peer review history of their article (what does this mean?). If published, this will include your full peer review and any attached files.

Reviewer #1: No

Reviewer #2: No

---

## [Editor Report · Acceptance letter]

23 Aug 2021

PONE-D-21-08855R1 

Coronary heart disease and stroke mortality trends in Brazil 2000-2018 

Dear Dr. Moreira:

I'm pleased to inform you that your manuscript has been deemed suitable for publication in PLOS ONE. Congratulations! Your manuscript is now with our production department. 

Kind regards, 

on behalf of

Dr. Venkata Naga Srikanth Garikipati 

Academic Editor

PLOS ONE